# Black Trumpet [*Craterellus cornucopioides* (L.) Pers.]—Bioactive Properties and Prospects for Application in Medicine and Production of Health-Promoting Food

**DOI:** 10.3390/nu16091325

**Published:** 2024-04-28

**Authors:** Iwona Adamska, Katarzyna Felisiak

**Affiliations:** Department of Fish, Plant and Gastronomy Technology, West Pomeranian University of Technology in Szczecin, 70-310 Szczecin, Poland; iwona.adamska@zut.edu.pl

**Keywords:** black trumpet, *Craterellus cornucopioides*, chemical composition, bioactivity, food production

## Abstract

Black trumpet (*Craterellus cornucopioides*) is a mushroom present in many countries but underestimated. The aim of this publication is to present the latest state of knowledge about the chemical composition and bioactivity of *C. cornucopioides* and the possibility of its application in food. According to researchers, black trumpet is very rich in nutritional compounds, including unsaturated fatty acids (mainly oleic and linoleic acids), β-glucans, minerals, and vitamins as well as polyphenols and tannins. It also contains compounds influencing the sensory properties, like free amino acids and nucleotides as well as sugars and polyols, mainly mannitol. Many of the described components show high nutritional and bioactive properties. Therefore, *C. cornucopioides* shows antioxidant activity and immunostimulating, anti-inflammatory, and anticancer effects as well as antibacterial, antifungal, antiviral, and antihyperglycemic effects. This makes black trumpet, also called horn of plenty, a mushroom with great potential for use both in medicine and directly in food. So far, black trumpet is not widely used in food, especially processed food. There are only a few studies on the use of dried black trumpet in sausages, but there is great potential for its use in food.

## 1. Introduction

Black trumpet (*Craterellus cornucopioides*) is an edible wild mushroom that is not very popular in Europe as a food ingredient. Its taste is usually known only to mushroom enthusiasts who are looking for new culinary experiences. Although its chemical composition is also relatively poorly known due to its bioactivity and the possibility of using it in medicine, this mushroom can be an element that changes the sensory value of dishes, and it can supplement the diet with health-promoting ingredients. In recent years, several works have appeared on the composition and functional properties of black trumpet. However, there are significant differences in the results presented by different authors. In turn, in the review of Bumbu et al. [1], the authors mainly presented the bioactive ingredients and biological properties of black trumpet without taking into account the basic composition or potential applications. Therefore, the aim of this publication is to present the latest state of knowledge about the chemical composition and bioactivity of *C. cornucopioides*, including differences between the results of different studies. Moreover, as it is a novelty, the possibilities of using this mushroom and the substances it contains in medicine and the production of functional food will be taken into account.

## 2. Characteristics and Occurrence

Black trumpet [*Craterellus cornucopioides* (L.) Pers], also called horn of plenty, black chanterelle, or trumpet of the dead, is a mushroom belonging to the phylum Basidiomycota, order *Cantharellales*. Its appearance is unusual for mushrooms: the mature fruiting body is very dark (dark brown to black) and has the shape of a slightly wrinkled, narrow funnel with a hollow center. The upper part of the fruiting body is rolled outwards (Figure 1). Fruiting bodies usually grow in scattered groups or clusters [2,3,4].

This mushroom occurs in deciduous and mixed forests. It belongs to the group of ectomycorrhizal organisms and enters into trophic systems with both deciduous trees (usually oaks and beeches) and conifers [2,3,4,5]. Its presence has been found in North America, Europe, and Asia [3,6] as well as South America and Australia [7]. Although it is a mushroom imported from Turkey [8], it still occurs only naturally (the technology for its cultivation has not been developed yet) [9]. 

## 3. Chemical Composition

### 3.1. Proximate Composition

The caloric value of the fruiting bodies of *Craterellus cornucopioides* was estimated at 248.0–413.2 kcal [10,11,12,13]. This is due, among other aspects, to the high water content (62.5–93.2% [11,12,13,14]) and low fat content (4.9–6.9% dw [10,11,12,13,14,15]). Only the study by Ouali et al. [16] showed a very high fat content (61.7% dw); however, they had used a different method of lipid content determination (gravimetrical from chloroform–methanol extracts instead of the Soxhlet method) than other researchers. Quali et al. [16] studied nine species of mushrooms and the lipid content in two of them was extremely high (>60%). In the case of all other studies, the crude lipid content was much lower than in other species. 

The protein content in these mushrooms varied greatly and ranged from 11.8% to 69.4% dw. Differences in protein content may be caused, among other aspects, by the correction factor used for the calculation of protein content from nitrogen determined by the Kjeldahl method: either 4.38 [12,13,16] or 6.25 [11,15]. According to research by Turfan et al. [17], *C. cornucopioides* contains the highest amount of total soluble protein (126.6 mg/g) among the 15 compared taxa (species and strains) of mushrooms. Large differences were also observed in the content of carbohydrates—their total content ranged from 6.2% to 45.6% dw. Differences in protein content and lipid content affected the calculated amounts of carbohydrates. The lowest value (6.2%) was demonstrated by Quali et al. [16], who found the highest lipid content. Among saccharides, sugars (glucose, fructose, sucrose) and sugar alcohols (mannitol) have to be mentioned as well as the very important nondigestible polysaccharide β-glucans, which are classified as dietary fiber [10,12,17]. 

The ash content ranged from 10.1% to 17.4% dw [10,11,12,14,15,16,18] (Figure 2). Differences in the amount of minerals in mushrooms occur frequently and depend strongly on the composition of the soil (the differences can be huge). Dimopolou et al. [15] showed that the fruiting bodies of *C. cornucopioides* contained 4.7% dw fibers and 0.1% dw salt. The share of crude fiber in the fruiting bodies examined by Odoh et al. [14] was as much as 16.1% dw. 

### 3.2. β-Glucans 

According to the analyses of Özcan and Ertan [19], β-glucans constituted 11.3% of the mass of black trumpet fruiting bodies, which was the lowest value among the five compared species (the most of these ingredients, 14.6%, was contained in the fruiting bodies of *Agaricus bisporus*). Similarly, Mirończuk-Chodakowska [20] showed that black trumpet fruiting bodies contained 15 g of β-glucan per 100 g dw, which was one of the lowest values among the compared mushroom species (the highest amount was found in *Tricholomopsis rutilans*—40.9 g/100 g dw). The total content of 1,3-1,6-β-D-glucan in *C. cornucopioides* was 4.5 g/100 g dw, which was also one of the lowest values compared to other tested species. The fruiting bodies of *Auricularia auricula-judae* contained the most of these substances—16.8 g/100 g dw [19]. According to the research of Guo et al. [21,22], black trumpet also contains triple helix polysaccharides. Radović et al. [13] found that thermal treatment (cooking of dried fruiting bodies of *C. cornucopioides*) caused a more than 10-fold decrease in the content of both α-glucans and β-glucans. The content of total glucans changed from 16.0 g/100 g dw to 1.5 g/100 g dw, and in the case of β-glucans, from 15.7 g/100 g dw to 1.4 g/100 g dw. 

### 3.3. Fatty Acids

Barros et al. [10] and Radović et al. [13] showed that among the fatty acids occurring in the fruiting bodies of *Craterellus cornucopioides*, unsaturated acids (UFA) predominated (accounting 75.9–83.6% of all fatty acids), the majority of which were monounsaturated fatty acids (MUFAs, accounting for almost 60% to 61.4%). Similar conclusions were reached by Dimopoulou et al. [15], who found the advantage of unsaturated fatty acids (69.6% of all fatty acids) over saturated acids, and by Ouali et al. [16] (UFA accounted 54.9% of all fatty acids). However, in the study by Ouali et al. [16], unlike in studies by other authors, polyunsaturated fatty acids (PUFAs) dominated among UFAs, accounting for 33.0%.

According to the research of Barros et al. [10], the most abundant fatty acids were oleic and linoleic acids (51.9% and 23.7%), followed by stearic and palmitic acids (7.9 and 6.7%). In the mushrooms studied by Radović et al. [13], oleic acid also dominated, accounting for 60.8% of all fatty acids, but stearic acids (12.4%), linoleic acid (10.8%), and palmitic acids (10.0%) also had a large share. Ouali et al. [16] found that among fatty acids, linoleic and oleic acids had the largest share (21.0% and 20.3%, respectively), followed by palmitic acid (13.9%), dihomo-γ-linolenic (10.9%), and margaric acid (8.9%). 

### 3.4. Mineral Composition

Research on the mineral composition showed large differences depending on the origin of the research material (mushrooms). The fruiting bodies of *Craterellus cornucopioides* analyzed by Vetter [18] were the richest in phosphorus and potassium among those compared in Table 1, the most copper and zinc were contained in mushrooms collected in Turkey [17], and those from Tunisia showed the highest content of magnesium and iron [16] (Table 1). According to research by Yildiz et al. [23] and Ouali et al. [16], these mushrooms may also be a good source of calcium.

### 3.5. Vitamins 

#### 3.5.1. Ascorbic Acid 

Barros et al. [10] and Liu et al. [24] showed a similar content of vitamin C in the fruiting bodies of *Craterellus cornucopioides* (0.87 mg/g and 0.81 mg/g dw, respectively), while Cağlarirmak [25] reported 1.89 mg/100 g wet weight (ww). According to Barros et al. [10], these mushrooms contained the most ascorbic acid among the compared mushrooms, although a similar content was found in the fruiting bodies of *Cantharellus cibarius* (0.86 mg/g dw). Vamanu and Nita [26] found that extracts from *C. cornucopioides* contained less ascorbic acid than extracts obtained from the M2191 strain of *Pleurotus ostreatus* mushrooms and from *Marasmius oreades* mushrooms (0.87 mg/100 g of extract, 2.71 mg/100 g of extract, and 2.62 mg/100 g of extract, respectively). 

#### 3.5.2. Tocopherols

The content of total tocopherols in *C. cornucopioides* is low compared to other species of tested mushrooms. Barros et al. [10] showed only 1.87 µg/g dw in black trumpet fruiting bodies, while in *Agaricus bisporus* it was 2.41 µg/g dw, in *A. silvicola*—3.23 µg/g dw, and in *Boletus edulis* as much as 10.65 µg/g dw. A similarly low content, 1.94 µg/g, was found by Liu et al. [24]. 

According to Barros et al. [10], the highest share of tocopherols in the black trumpet was β-tocopherol (1.55 µg/g dw), while α-tocopherol and γ-tocopherol contents were much smaller (0.24 µg/g dw and 0.08 µg/g dw, respectively). Similarly, in the study by Liu et al. [24], α-tocopherol dominated quantitatively (1.15 µg/g dw), while γ-tocopherol and δ-tocopherol had a small share in the total tocopherol content (0.62 µg/g dw and 0.17 µg/g dw, respectively). Fruiting bodies of *C. cornucopioides* studied by Radović et al. [13] contained 0.479 mg/g dw of α-tocopherol. Barros et al. [10] found a greater amount of α-tocopherol in *Agaricus silvicola* (1.30 µg/g dw), *A. bisporus* (0.75 µg/g dw), and *A. silvaticus* (0.49 µg/g dw). For β-tocopherol, the values were *Boletus edulis* (8.90 µg/g dw), *A. silvicola* (1.93 µg/g dw), and *A. bisporus* (1.66 µg/g dw), and for γ-tocopherol, the values were *B. edulis* (1.42 µg/g dw), *Marasimus oreades* (1.30 µg/g dw), and *Calocybe gambosa* (0.14 µg/g dw).

Extraction using a fluidized bed allowed researchers to obtain extracts containing 117.4 mg of α-tocopherol per 100 g of extract (the most among the compared seven taxa of fungi) and 6.42 mg/100 g of extract γ-tocopherol (the least among the tested species, although its occurrence was not found in four taxa) [26]. Research conducted by Radović et al. [13] showed the presence of vitamin E only in cyclohexane extract in an amount of 2816 mg/100 g of dry extract, while water and methanol extracts did not contain it.

#### 3.5.3. Other Vitamins 

Liu et al. [24] showed the presence of ergosterol in *C. cornucopioides*, a precursor of vitamin D, in the amount of 3.27 mg/g dw; however, in the study by Gil-Ramirez et al. [27], it was much less, only 0.79 mg/g dw. Villares et al. [28] also reported low content of ergosterol in black trumpet, 0.44 mg/g, so five among eight studied mushrooms contained much more ergosterol. According to Despatliev et al. [29], ergosterol accounted for 72.8% of all sterols present in *C. cornucopioides*, while in *Cantharellus cibarius* it was only 42.4%. Generally, when mushrooms are exposed to UV radiation, ergosterol is transformed into a pro-vitamin and then to ergocalciferol, vitamin D2 [30]. However, the levels of ergocalciferol in mushrooms are usually low and both compounds are reduced by cooking processes [31,32]. Cyclohexane extract prepared from the fruiting bodies of *C. cornucopioides* also contained vitamin D3 (cholecalciferol) in the amount of 89.3 mg/100 g of dry extract (which corresponded to 1.52 mg/100 g dw of mushroom) and vitamin A (16.1 mg/100 g dry extract); however, the first result was not supported by any publication [13]. Radović et al. [13] found that methanol and water extracts also contained vitamins B1, B2, B3, and B6. There was more vitamin B1 in water extract than in methanol extract (54.7 and 19.4 mg/100 g of dry extract, respectively). The remaining vitamins were present in greater amounts in methanol extracts than in water extracts. The greatest difference was found in the case of vitamins B3 (367.0 and 249.0 mg/100 g dry extract, respectively) and B6 (17.0 and 8.3 mg/100 g dry extract). Cağlarirmak [25] found the presence of vitamins B1, B2, B3, B6, and B9 in the fruiting bodies of this species in amounts 0.11, 0.06, 3.34, 0.86, and 17.83 mg/100 g ww, respectively. Watanabe et al. [33] showed that black trumpet fruiting bodies from four regions in Europe contain from 1.79 to 2.65 µg/100 g dw of vitamin B12. Compared to five other mushroom species, these were the highest values. 

##### 3.5.4. β-Carotene

Barros et al. [10] showed that the fruiting bodies of *C. cornucopioides* were very rich in β-carotene (containing 12.77 µg/g). The higher amount occurred only in *Cantharellus cibarius* (13.56 µg/g), and the lowest in *Agaricus bisporus* (1.95 µg/g). Vamanu and Nita [26] found that the ethanol extract prepared from *C. cornucopioides* contained 0.14 mg/100 g of β-carotene. A higher concentration was found only in the extract prepared from the *Tuber melanosporum* (0.18 mg/100 g of extract) and the lowest in the extract from *Agaricus bisporus* (0.002 mg/100 g of extract). Kol et al. [34] showed that methanol extract contained 6.34 µg/mg of β-carotene and water extract 3.89 µg/mg. 

### 3.6. Lycopene

According to the research of Barros et al. [10], the fruiting bodies of *C. cornucopioides* were the richest in lycopene among the eight compared species and contained 5.13 µg/g of extract. This value was 0.07 µg/g higher than the content found in the fruiting bodies of *Cantharellus cibarius*, which are considered a rich source of this ingredient. For comparison, the lowest lycopene content was found in *Agaricus bisporus* (0.91 µg/g of extract) [10]. Vamanu and Nita [26] also showed that ethanol extracts of *C. cornucopioides* contained lycopene, but it was less than in one of the tested strains of *Pleurotus ostreatus* and less than in *Tuber melanosporum* (0.07 mg/100 g of extract in *C. cornucopioides*, 0.12 mg/100 g of extract in *P. ostreatus,* and 0.1 mg/100 g of extract in *T. melanosporum*). Kol et al. [34] showed that depending on the solvent, the lycopene content in the extracts of *C. cornucopioides* varied, and it was 2.49 µg/mg for water extract and 5.55 µg/mg for methanol extract.

### 3.7. Phenolic Compounds, Flavonoids, and Tannins

Most studies found that the fruiting bodies of *C. cornucopioides* contained small amounts of total phenolic compounds, from 2.13 mg/g dw [10] to about 5 mg/g dw [16]; however, Turfan et al. [17] found as much as 37.5 mg/g dw. In research conducted by Barros et al. [10], among the eight compared species, the largest amount of these substances was found in the fruiting bodies of *Agaricus silvaticus* (8.94 mg/g dw), and smaller amounts than that recorded in *C. cornucopioides* were found only in *Calocybe gambosa* (1.7 mg/g dw) and *Cantherellus cibarius* (0.88 mg/g dw). Among the nine species compared by Ouali et al. [16], the highest total phenolic compounds were found in *Hericium erinaceus* (above 11 mg/g dw), and among the 15 taxa studied by Turfan et al. [17] in *Boletus edulis* (157.4 mg/g dw). A very low content of total phenolic compounds, amounting to 1.6 mg GA/g dw in the fruiting bodies of *C. cornucopioides,* was also demonstrated by Dospatiev et al. [35]. Vamanu and Nita [26] found that the extract obtained from the fruiting bodies of *C. cornucopioides* contained total polyphenols at the level of 88.4 mg GA/100 g of extract, which was one of the average values among the seven compared mushroom species. The largest amount was contained in the fruiting bodies of *Tuber melanosporum* (122.4 mg of gallic acid/100 g of extract). According to research by Radović et al. [13], the total phenolic content in the water extract was almost twice as high compared to the methanol extract (17.0 and 8.9 mg GA/g extract, respectively). 

Both Barros et al. [10], Turfan et al. [17], and Ouali et al. [16] showed that the fruiting bodies of *C. cornucopioides* contained small amounts of flavonoids (1.71 mg/g, 8.8 mg/g, and about 5 mg/g dw, respectively). The largest amount of these substances Barros et al. [10] found in *Agaricus silvaticus* (3.4 mg/g), Turfan et al. [17] in *Ganoderma lucidum* (30.7 mg/g), and Ouali et al. [16] in *Ramaria flavescens* (above 17 mg/g dw). Similarly, Vamanu and Nita [26] found that the extract prepared from *C. cornucopioides* contained only 151.5 µg of quercetin/100 g of extracted flavonoids, while the largest amount among the seven fungal taxa tested was contained in the extract from *Tuber melanosporum* (414.0 µg of quercitin/100 g of extract). 

Similar observations regarding the low content of total phenolic compounds and total flavonoids in black trumpet fruiting bodies were made by Palacios et al. [36]. Their studies showed that ferulic, gallic, *p*-hydroxybenzoic, homogentisic and protocatechuic acids, myricetin, and pyrogallol were present in methanol extracts. 

Fruiting bodies of *C. cornucopioides* contained approximately 20 mg/g dry weight of tannin compounds, which was the average value among nine fungal taxa studied by Ouali et al. [16], while *Lactarius deliciosus* contained the most of these substances (above 25 mg/g).

### 3.8. Ingredients Influencing the Sensory Properties of Black Trumpet

Although raw fruiting bodies of *C. cornucopioides* have a mild taste and pleasant smell [3], they should be eaten after heat treatment. Properly prepared, they are tasty and delicate in taste and also have a pleasant smell [4]. They are considered the most aromatic among other edible mushrooms from the order *Cantherellales* [37]. Both fresh and soaked, they are flexible after drying [4]. 

The taste of mushrooms is related to the content and composition of non-volatile components, including the combination of free amino acids and nucleotides characteristic of a given species [12,17,38,39,40,41] and soluble sugars and polyols. Turfan et al. [17] and Beluhan and Ragnogajec [12] believe that the relatively high content of sugars and polyols, which affects the sweetish taste of mushrooms, is a feature particularly desired by consumers (Figure 3).

#### 3.8.1. Sugars and Polyols

Fruit bodies of *Craterellus cornucopioides* contain from 10.8 to 15.2 g/100 g dw total soluble carbohydrates [10,12,16] (Figure 4). However, according to a comparison conducted by Beluhan and Ragnogajec [12], these mushrooms are the poorest in these ingredients among the 10 edible species compared. 

Turfan et al. [17] showed the presence of glucose (51.6 mg/g; it constituted 15.0% of all soluble sugars and polyols), fructose (9.2 mg/g; 2.7%), and sucrose (1.8 mg/g; 0.5%) in the fruiting bodies of *C. cornucopioides*. On the other hand, the presence of maltose and melezitose [10] and mannose [12] has not been found in them so far, although the presence of these substances has been demonstrated in other species of mushrooms. 

Compared to other mushroom species, the fruiting bodies of *C. cornucopioides* were among the richest in terms of mannitol content (a higher amount was found only in *Agaricus campestris*), but the poorest in trehalose and glucose [12]. 

#### 3.8.2. Amino Acids and Free Nucleotides

Amino acids found in food products differ in their solubility in various solvents and the taste sensation they produce. Some of them (proline, hydroxyproline, alanine, and glycine) easily dissolve in water, and depending on the pH, they give the products a sweet (all four amino acids) or bitter (proline and hydroxyproline) taste. The threshold values for the perception of taste sensation for most of them are very low and are 3.0, 0.5, 0.6, and 1.3 mg/cm3, respectively. The solubility in water of other amino acids giving sweet (e.g., serine), sour (e.g., aspartic acid and glutamic acid) or bitter tastes (phenylalanine, isoleucine, leucine, methionine, tryptophan, and valine) is weaker [42,45,46]. Monosodium glutamate, sodium aspartate, glutamic acid, and 5′-nucleotides, responsible for the umami taste, play a special role in food [12,38,42,47]. To a lesser extent, aspartic acid, glutamine, serine, and methionine are also responsible for the perception of this taste [42]. 

Turfan et al. [17] found that the fruiting bodies of *C. cornucopioides* contained one of the highest values of the total amount of free amino acids among the 15 compared fungal taxa, i.e., 6.6 mg/g (the highest value was found in *Marasmius oreades*—7.6 mg/g). According to research by Dospatliev et al. [40], dry fruit bodies contained an average of 30.4 mg amino acids /kg dw, the largest shares of which were glutamine (11.9 mg/kg dw), arginine (4.1 mg/kg dw), ornithine (3.2 mg/kg dw), glutamic acid (2.8 mg/kg dw), and serine (1.6 mg/kg dw). The remaining amino acids had a smaller contribution (Figure 5). The dominant group of amino acids found in these mushrooms is mostly responsible for the sweet, sour, and umami tastes (glutamine, serine, ornithine, and glutamic acid) [42,43], and only arginine is associated with a bitter taste. Research by Radović et al. [13] showed that the largest share in the fruiting bodies of *C. cornucopioides* were amino acids, which, according to the classification of Beluhan and Ragnogajec [12], are responsible for the bitter taste. They constituted 45.1% of all amino acids found in these mushrooms (3.1 mg/g dw). These mushrooms contained the most arginine (32.1% of all amino acids; 2.2 mg/g dw) and glutamic acid (20.1%, 1.4 mg/g dw), responsible for the bitter and monosodium glutamate-like tastes, respectively.

Beluhan and Ragnogajec [12] found that in the fruiting bodies of *C. cornucopioides* collected in the forests of Croatia, the amino acid determining the umami taste (glutamic acid) was quantitatively dominant. It constituted 70.0% of all amino acids present in these mushrooms. Amino acids responsible for bitter and sweet tastes (12.0% and 10.8%, respectively) and tasteless amino acids (7.2%) had a much smaller share. Amino acids with a significant share were lysine (accounting for 7.0% of all amino acids), threonine (6.8%), histidine (5.4%), and alanine (3.5%). The remaining amino acids were present in small amounts. On the other hand, according to the research of Beluhan and Ragnogajec [12], the fruiting bodies of *C. cornucopioides* contained the highest amount of free 5′-nucleotides among other tested fungi (35.4 mg/g dw), which included 5′-adenosine monophosphate (5′-AMP), 5′-inosine monophosphate (5′-IMP), 5′-xanthosine monophosphate (5′-XMP), 5′-cytosine monophosphate (5′-CMP), 5′-guanosine monophosphate (5′-GMP), and 5′-uridine monophosphate (5′-UMP). These ingredients determine the taste of the product. A special role is played by 5′-GMP, which enhances the flavor and at the same time gives the products a meaty taste [31]. However, the sum of the 5′-GMP, 5′-IMP, and 5′-XMP content is considered to be the factor that demonstrates the intensity of the product taste [48]. According to Beluhan and Ragnogajec [12], this coefficient for *C. cornucopioides* was the highest among other compared fungi and amounted to 13.9 mg/g dw, while for the remaining species it ranged from 0.4 to 5.3 mg/g dw. However, the EUC index, which is an indicator of the umami taste as a result of the synergism of monosodium glutamate-like (MSG-like) components and free 5′-nucleotides, in the case of *C. cornucopioides*, was one of the lowest among the compared species and amounted to only 120.9 g MSG/100 g dw (in the case of *Boletus edulis*, this index reached a value of as much as 1186.5 g MSG/100 g dw). In the study by Radović et al. [13], the largest share of 5′-UMP was demonstrated (1.6 mg/g dw, which accounted for 38.8% of all 5′-nucleotides), and the other three nucleotides had a smaller share (5′-AMP: 0.9 mg/g dw, 21.8%; 5′-GMP: 0.8 mg/g dw, 20.8%; 5′-CMP: 0.7 mg/g dw, 18.5%). 

Fons et al. [44] found that the volatile components responsible for the sensory properties of *C. cornucopioides* fruit bodies were mainly oct-1-en-3-ol, limonene, and (E)-Oct-2-enol constituting, respectively, 17.1%, 14.8%, and 12.1% of the volatile compounds present in these mushrooms. 

### 3.9. Other Ingredients

Liu et al. [49], examining the chemical composition of *C. cornucopioides* culture broth, isolated three substances in the form of colorless oils: 4-oxohex-5-enyl acetate (C_8_H_12_O_3_Na), 4-oxohex-1,6-diyl diacetate (C_10_H_17_O_5_), and 6-hydroxy-4-oxohexyl acetate (C_8_H_12_O_3_Na), belonging to the group of keto esters. Additionally, eight components belonging to the sesquiterpenoid group were isolated from black trumpet cultures: gymnomitr-3-en-10β,15-diol, illudalenol, illudin F, illudin M, illudin T, and craterellins A-C [50].

## 4. Bioactivity of Black Trumpet Fruiting Bodies

### 4.1. Antioxidant Activity

The antioxidant activity of black trumpet has been studied by a few authors, but the results differed depending on the extraction method, the testing method, and the expression of the results (Table 2). 

According to Queirós et al. [51], *Craterellus cornucopioides* was characterized by average antioxidant activity, which was the third highest among five species examined. The concentration of dry extract necessary to inhibit β-carotene bleaching was low, 1.70 mg/mL, so the fruiting bodies of black trumpet compounds were effective to prevent lipid oxidation. In case of DPPH scavenging activity and reducing power, the effective concentrations causing 50% of antioxidant activity (EC_50_) were lower than 7.5 mg/mL, so they were considered by the authors as low (high antioxidant activity). However, according to Vasdekis et al. [52], *C. cornucopioides’* ability to scavenge DPPH free radicals, expressed as EC_50_ of dry extract, was higher than 40 mg/mL and did not differ compared to other fungi from the *Cantharellaceae* family (*Cantharellus cibarius* and *Cantharellus cinereus*) and was similar to another 14 fungal species among the 29 studied. The highest antioxidative properties were found in *Amanita citrina* (4.0 ± 0.0 mg/mL), *Ganoderma lucidum* (4.0 ± 0.0 mg/mL), and *Agaricus urinascens* (4.9 ± 0.1 mg/mL). There was clear dependency between antioxidant activity and total polyphenol content; *C. cornucopioides* was in the group of four species with the lowest polyphenol content, although not in terms of flavonoids [52]. 

Mešić et al. [53], examining the relationship between the composition and morphological features of 16 species of wild mushrooms (in 23 samples), showed that *C. cornucopioides* were characterized by the third highest DPPH scavenging ability (31 µmol Trolox equivalent TE/g dw) after *Psathyrella piluliformis* (40 µmol TE/g dw) and *Lycoperdon perlatum* (35 µmol TE/g dw). It showed the fifth highest ferric reducing power (31 µmol Fe^3+^/g dw), and *P. piluliformis* also showed the highest FRAP values of 70 and 52 µmol TE/g dw (in two independent tested samples). 

In turn, Costea et al. [54] investigated the antioxidant activity of *C. cornucopioides*, depending on the type of extracts. They found that water dry extract was characterized by the highest ability to scavenge DPPH and ABTS free radicals as well as ferric reducing power, expressed as ascorbic acid equivalents per 1 g of dry extract (AAE/g), while methanolic dry extract showed the highest chelating activity, expressed as Na2-EDTA equivalents per one gram of dry extract. Ethanolic extracts showed the lowest antioxidant properties. Comparison of the effect of dry extract concentration (0.2–1.8 g/mL) on 50% inhibition of absorbance (IC_50_) confirmed that water dry extracts caused higher inhibition of DPPH and ABTS free radicals than other extracts, and ethanol extracts showed the lowest inhibition and the lowest absorbance. In turn, methanol extracts showed the highest inhibition in chelating activity. This was due to the content of polyphenols in these extracts; in the water extract, they were more than twice as high as in the ethanol and methanol extracts [54]. 

Kosanić et al. [56] showed that acetone extract of black trumpet was characterized by antioxidant activity expressed as DPPH radical scavenging activity, and the IC_50_ was 19.7 ± 1.1 µg/mL, while ascorbic acid as positive control was 6.41 ± 0.2 µg/mL. Superoxide anion scavenging IC_50_, 221.8 ± 3.1 µg/mL, was two times greater than ascorbic acid. However, ferric reducing power was very low. 

Generally, it can be said that the black trumpet is characterized by relatively low DPPH scavenging activity, but its antioxidant activity against ABTS cation radicals is high. 

Ferric reducing power was rather low [54], especially in acetone extract [56]; however, copper reducing power was three times lower than that of α-tocopherol [55]. Extracts were efficient as ferrous chelating agents [54] and agents preventing lipid oxidation determined as inhibition of β-carotene bleaching [51] and linoleic acid oxidation [36,55]. 

The antioxidant activity of mushrooms is related to the presence of polyphenols, flavonoids, tocopherols, etc., as well as polysaccharides. Yang et al. [57] showed that methanolic extract of polysaccharides extracted from black trumpet fruiting bodies showed high antioxidant activity against DPPH and ABTS radicals (0.10 and 0.15 mg/mL, respectively). 

### 4.2. Immunostimulating and Anti-Inflammatory Effects

Research conducted in Tanzania has shown that a preparation consisting of the powdered fruiting bodies of several species of mushrooms, including *C. cornucopioides*, effectively strengthens immunity in HIV-infected patients [58].

According to the research of Guo et al. [21,22,59], polysaccharides occurring in the fruiting bodies of this mushroom species, especially polysaccharides characterized by a triple helix, may be responsible for the extremely beneficial immunostimulating effect. They have the ability to stimulate the functioning of the immune system in mice by stimulating the activity of RAW264.7 macrophages. Ding et al. [60] came to similar conclusions as a result of examining the bioactivity of the CC-M polysaccharide isolated from *C. cornucopioides*. It increased the activity of T and B lymphocytes and RAW264.7 macrocytes and stimulated the signaling pathways of MAPK, PI3K-Akt, NF-κB, and NOD-like receptor signaling pathway receptors as a result of stimulating the secretion of CD163, IL-1β, IL-6, IL-10, and TNF-α. Also, Xu et al. [61] found that CCPP-1 polysaccharides obtained from *C. cornucopioides* can affect the NF-kB signaling pathway and alleviate inflammation by reducing the amount of pro-inflammatory cytokines IL-1β, IL-18, and TNF-α and the mediator iNOS and by reducing the amount of ROS and NO in cells. Moreover, CCPP-1 polysaccharides had the ability to restore GPx and CAT levels to appropriate levels and inhibit the formation of MDA and reactive oxygen species (ROS) inside cells. This action helps protect erythrocytes against changes caused by oxidative stress (hemolysis) [57]. 

Zhang et al. [62] found that the polysaccharide CCP2 with a catenarian pyranose structure, consisting of galactose, glucose, mannose, and xylose, isolated from *C. cornucopioides*, also exhibited immunomodulatory effects in in vitro and in vivo studies. It activated the increased production of cytokines IL-2, IL6, and IL-8, TLR4 protein, and protein kinases NF-κB p 65, TRAF6, and TRIF. This stimulated the activity of macrophages and immune processes in cells. 

The effectiveness of *C. cornucopioides* extract in preventing or reducing inflammation by reducing the expression of NO and IL-6 was also previously demonstrated by O’Callaghan et al. [63]. Moro et al. [64] showed that the methanol extract from this mushroom only inhibits the production of NO and the expression of iNOS, without causing changes in the levels of IL-1b and IL-6 (Figure 6).

### 4.3. Antibacterial, Antifungal, and Antiviral Effects

It was shown that methanol and acetone extracts obtained from *C. cornucopioides* inhibited the development of *Staphylococcus aureus*. Moreover, they were characterized by antibacterial activity against *Klebsiella pneumoniae*, but this effect was observed only at the highest tested concentration (200 mg/mL) [19]. Kosanić and colleagues [56] found that acetone extracts affect not only *Staphyllococcus aureus* but also inhibit the activity of *Bacillus cereus*, *B. subtilis*, *Escherichia coli*, and *Proteus mirabilis*. Kol et al. [34] demonstrated strong antibacterial effectiveness of methanol and water extracts against *Agrobacterium tumefaciens*, *Bacillus licheniformis*, *B. subtilis*, *Enterococcus faecalis*, *Escherichia coli*, and *Staphylococcus aureus* (ATCC 2921). Acetone extracts also limited the activity of the fungi *Aspergillus niger*, *Candida albicans*, *Mucor mucedo*, *Penicillium italicum,* and *Trichoderma viride* [56]. 

It has also been shown that the CCP polysaccharide isolated from the fruiting bodies of *C. cornucopioides* influences the species and quantitative composition of microorganisms constituting the intestinal flora and their activity, promoting species that have a beneficial effect on digestive processes. The authors of the study related the research results to the possibility of using CCP in the prevention of digestive system diseases [65]. 

The literature also contains information about the antiviral effect of aqueous extracts of this fungus against *Vaccinia virus* [66]. 

### 4.4. Anticancer Effect

In order to examine the anticancer activity of *C. cornucopioides*, the following were used: extracts obtained from the fruiting bodies of the fungus using various solvents (water, methanol, acetone, cyclohexane, or dichloromethane), selected substances isolated from liquid extracts, and powdered dried mushrooms (a mixture of several species) (Figure 7).

Kol et al. [34] showed that both aqueous and methanol extracts of *C. cornucopioides* had the ability to inhibit the growth of HepG2 tumor cells (human hepatocellular cancer), but this effect was more pronounced in the case of methanol extracts (IC_50_ for the methanol extract was 3.14 mg/mL and for the water extract 18.41 mg/mL). It was observed that the anticancer effectiveness of both types of extracts increased with the increase in the applied concentration. Also, Vasdekis et al. [52] found that the methanol extract showed very strong anticancer activity against the A549 cancer cell line (adenocarcinomic human alveolar basal epithelial cells; IC_50_ < 1 mg/mL); moreover, this activity was significantly higher compared to other tested fungal species.

The acetone extract, however, showed a much greater effect on the HeLa cell line (human epithelial cervical cancer cells; IC_50_ 65.5 μg/mL) than on A549 (IC_50_ 108.2 μg/mL) and LS174 (human colorectal cancer cells; IC_50_ 131.7 μg/mL) [56]. Testing the effect of four types of extracts (aqueous, methanol, cyclohexane, and dichloromethane) on the A549, HeLa, and LS174 cancer cell lines showed that cyclohexane and dichloromethane extracts have anticancer activity against all tested cell lines. Cyclohexane extract had a strong effect on HeLa (IC_50_ 78.3 µg/mL) and a moderate effect on LS174 (IC_50_ 139.1 µg/mL) and on A549 (IC_50_ 141.9 µg/mL). Dichloromethane extract had a similar moderate effect on HeLa and LS174 (IC_50_ 135.6 and 135.7 µg/mL, respectively) and a weaker effect on A549 (IC_50_ 153.2 µg/mL). Aqueous and methanolic extracts showed only low cytotoxic activity towards LS174 (IC_50_ 191.5 and 186.6 µg/mL, respectively) [13]. 

The anticancer activity of eight sesquiterpenoids isolated from *C. cornucopioides* fruiting bodies against five cancer cell lines was also tested, but only one of them—craterellin C (C_15_H_22_O_4_)—showed moderate toxicity towards A549 cells (IC_50_ 21.0 μM) [50]. 

The health-promoting effects of mushrooms, including their anticancer effects, should be associated with the polysaccharides they contain, including beta-glucans [67,68]. However, according to comparative studies by Mirończuk-Chodakowska et al. [20], black trumpet fruiting bodies are not the richest in these substances. Moreover, according to the literature reports, GACOCA powder, obtained after grinding several species of dried mushrooms (*Cantharellus cibarius*, *C. cornucopioides*, *C. isabellinus*, *Fomes fomentarius*, *Ganoderma applanatum*, *G. lucidum*, *G. pfefferi*, *Phellinus igniarus*, *Schizophyllum commune*, and *Trametes versicolor*), has been successfully used in Tanzania as a therapeutic agent in the treatment of Kaposi’s sarcoma (skin cancer affecting patients with HIV/AIDS) [58]. 

### 4.5. Antihyperglycemic Effect

The ethanol extract from *C. cornucopioides* shows high α-glucosidase inhibitory activity (EC50 value 8.28 lg/mL) [24,69]. 

### 4.6. Other Activity

The methanol–water extract obtained from *C. cornucopioides* showed very weak inhibition of fat absorption as a pancreatic lipase inhibitor. Although the activity of the extract determined using the enzymatic kit was 95.6%, it increased significantly using the in vitro digestion model and amounted to as much as 181.0% [70].

## 5. Possibilities of Using Black Trumpet Fruiting Bodies in Food Production

Black trumpet fruiting bodies have a scent reminiscent of apricots, which is why they are appreciated by consumers [9]. On the one hand, their appearance is considered to be unappetizing due to its blackening during cooking [71], but according to Yamada [9], the natural shape of a black funnel is an additional stimulator of sensory experiences when consuming dishes with this mushroom. 

Fruiting bodies of *C. cornucopioides* are most often used in dried form (as raw material for preparing various dishes) or dried and ground into mushroom flour (for preparing soups) [8]. The latter method of being used as a seasoning for soups and sauces was also mentioned in the Lexicon of Mushrooms [71]. In Denmark, *C. cornucopioides* is sold in two forms: fresh and dried [5]. 

The antioxidant and antibacterial effects of the black trumpet fruiting bodies, demonstrated in many studies, led to the study of the effect of the fruiting bodies of this mushroom together with two other species (*Boletus edulis* and *Cantharellus cibarius*) as an addition to frankfurters. All these mushroom species significantly influenced the texture of the sausages. An improvement (increase) in the hardness and chewiness of the final product was achieved due to the higher protein content, which created a denser matrix compared to frankfurters without the addition of mushrooms. However, in this study, samples with the addition of *C. cornucopioides* obtained lower scores than the other two species of mushrooms. What was significant in the tests was the very low color rating of the product with the addition of black trumpet. After adding this mushroom, an immediate deterioration in color was observed, which disqualified the final product after 2 months of storage. Frankfurters with the addition of the other two species received much better ratings here [72]. 

Due to its dark color, black trumpet can be a substituted for mun mushrooms (*Auricularia auricula-judae*), used as one of the main ingredients in Asian (mainly Chinese) dishes. Moreover, this mushroom can be used in the production of, e.g., pasta, and it can be a valuable ingredient in snacks and crackers. It can be also a valuable ingredient in vegan “blood sausage”.

## 6. Conclusions

Black trumpet, also called horn of plenty, contains a high amount of water; however, the composition of dry matter varies depending on the world region and study. It was found that *C. cornucopioides* contains, among other components, unsaturated fatty acids (mainly oleic and linoleic acids), β-glucans, minerals, and vitamins as well as polyphenols and tannins. It also contains compounds influencing the sensory properties, like free amino acids and nucleotides as well as sugars and polyols, mainly mannitol. The presence of bioactive compounds causes *C. cornucopioides* to show antioxidant activity and immunostimulating, anti-inflammatory, and anticancer properties. It also exhibits antibacterial, antifungal, antiviral, and antihyperglycemic effects. Although black trumpet has been applied in traditional medicine and it is consumed in freshly prepared dishes, it has not yet been used in ready-to-eat processed products. So far, a few works have presented its use only in sausages. However, the use of fresh or dried mushrooms in food can provide important health benefits and could therefore be used in new, interesting food products.

## Figures and Tables

**Figure 1 nutrients-16-01325-f001:**
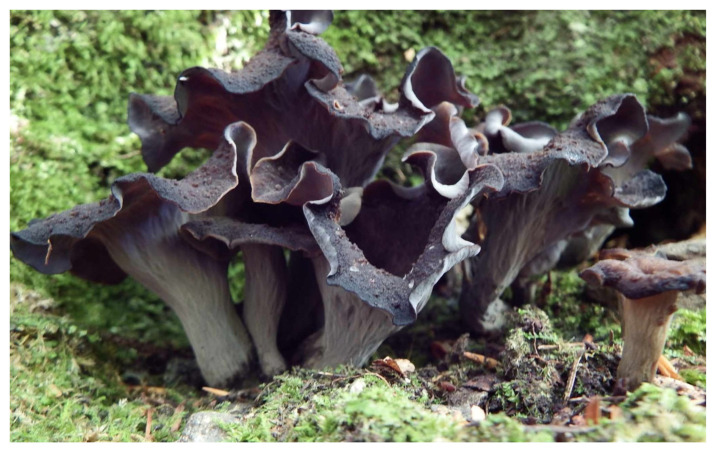
Fruiting bodies of the black trumpet (*Craterellus cornucopioides*). Photo: Pierino Bigoni.

**Figure 2 nutrients-16-01325-f002:**
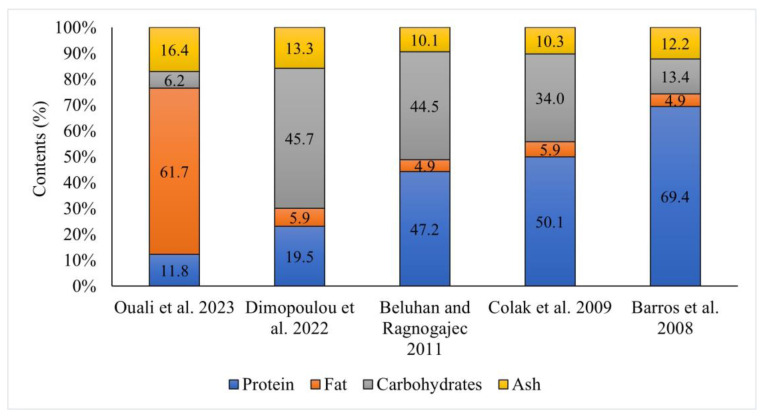
Basic composition of fruiting bodies of *Craterellus cornucopioides* (dry matter) [10,11,12,15,16].

**Figure 3 nutrients-16-01325-f003:**
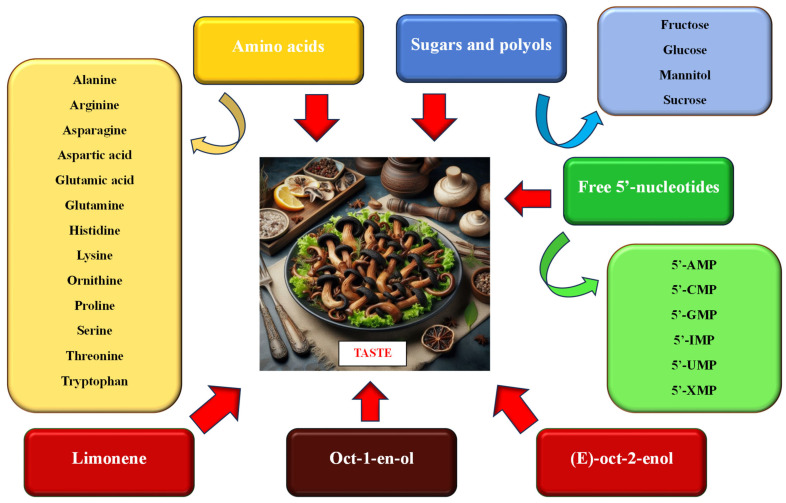
Chemical ingredients affecting the taste of food products containing black trumpet fruiting bodies (data sources: [10,12,13,16,17,40,42,43,44]; the figure was prepared by the authors, and the picture was generated using artificial intelligence in Bing AI).

**Figure 4 nutrients-16-01325-f004:**
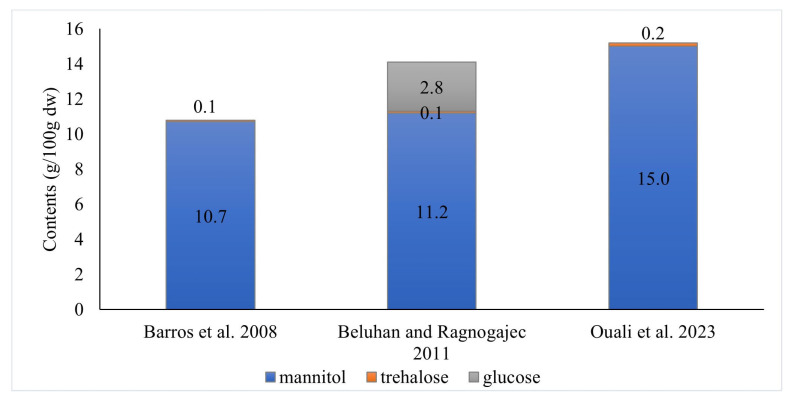
Composition of soluble sugars and polyols in fruiting bodies of *Craterellus cornucopioides* according to different authors [10,12,16].

**Figure 5 nutrients-16-01325-f005:**
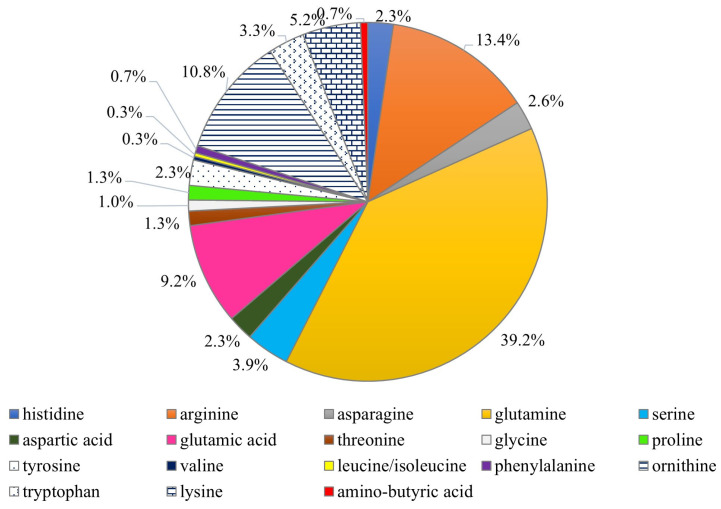
The percentage of free amino acids present in the fruiting bodies of *Craterellus cornucopioides* (data from: Dospatliev et al. [40]).

**Figure 6 nutrients-16-01325-f006:**
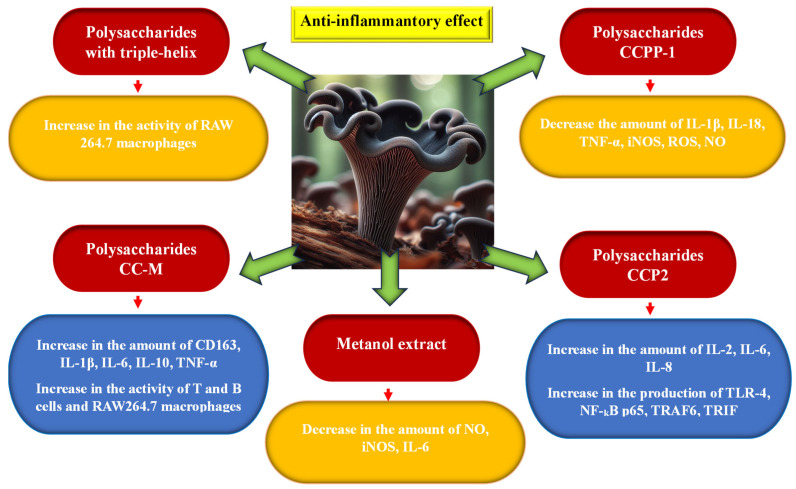
Anti-inflammatory activity of *Craterellus cornucopioides* fruiting bodies (data sources: [21,22,57,59,60,61,62,63,64]; the figure was prepared by the authors, and the picture was generated using artificial intelligence in Bing AI).

**Figure 7 nutrients-16-01325-f007:**
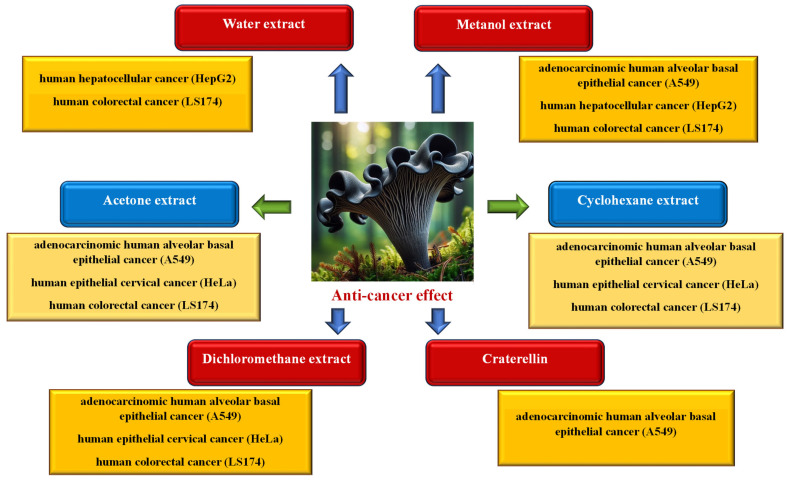
Anticancer activity of *Craterellus cornucopioides* fruiting bodies (data sources: [13,34,50,52,56]; the figure was prepared by the authors, and the picture was generated using artificial intelligence in Bing AI).

**Table 1 nutrients-16-01325-t001:** Mineral composition of fruiting bodies of *Craterellus cornucopioides* according to different authors [14,15,16,17,18,23].

Minerals	Ouali et al. [16] mg/kg dw	Dimopoulou et al. [15] mg/kg dw	Yildiz et al. [23] mg/kg dw	Turfan et al. [17] mg/kg dw	Odoh et al. [14] mg/g dw	Vetter [18] mg/kg dw
K	35,800.0	Nt	Nt	5471.9	10.02	37,100.0
P	2500.0	Nt	Nt	1462.4	Nt	6300.0
Mg	1000.0	978.0	151.6	17.6	10.82	Nt
Ca	800.0	Nt	935.5	163.6	20.89	240.0
Na	400.0	Nt	Nt	10.0	5.00	Nt
Fe	600.0	413.0	255.3	598.3	1.32	Nt
Cu	20.0	43.0	40.5	108.5	2.37	Nt
Zn	50.0	61.0	91.5	378.5	40.19	Nt
Mn	T	Nt	13.7	147.6	0.20	Nt
Cr	T	Nt	0.142	11.01	Nt	Nt
Se	Nt	14.0	3.08	Nd	Nt	Nt
Al	Nt	Nt	56.1	177.54	Nt	Nt

T—trace amount, Nt—not tested, Nd—not detected.

**Table 2 nutrients-16-01325-t002:** Antioxidant activities of black trumpet (*Craterellus cornucopioides*) extracts and extracted black trumpet polysaccharides (EC_50_—median effective concentration, causing 50% of antioxidant activity; IC_50_—concentration causing 50% inhibition of absorbance; AAE—ascorbic acid equivalents/g; TE—Trolox equivalent; Eq—equivalents).

Assay	Extract	Antioxidant Activity	Source
DPPH free radicals scavenging ability (DPPH)	Methanol extract	EC_50_ 6.33 ± 0.48 mg/mL Revealed antioxidant properties EC_50_ < 7.5 mg/mL	[51]
		EC_50_ > 40 mg dry extract/mL Ascorbic acid 0.1 mg/mL	[52]
		31 µmol TE/g d.w.	[53]
		20.1487 ± 0.7872 AAE/g	[54]
		IC_50_ 1.8 mg/mL	[54]
		EC_50_ 8.65 mg/mL	[55]
	Ethanol extract	18.105 ± 0.744 AAE/g	[56]
		IC_50_ >> 1.8 mg/mL	[54]
		EC_50_ 40.64 mg/mL	[24]
	Water extract	27.6787 ± 0.8780 AAE/g	[54]
		IC_50_ 1.0 mg/mL	[54]
		EC_50_ 8.65 mg/mL	[55]
		EC_50_ 26.37 mg/mL	[24]
	Acetone extract	IC_50_ 19.7 ± 1.1 µg/mL Ascorbic acid 6.41 ± 0.2 µg/mL	[56]
ABTS scavenging ability (ABTS)	Methanol extract	273.13 ± 0.5861 AAE/g	[54]
		IC_50_ 0.1 mg/mL	[54]
	Ethanol extract	295.34 ± 0.4906 AAE/g	[54]
		IC_50_ 0.2 mg/mL	[54]
	Water extract	559.29 ± 0.1967 AAE/g	[54]
		IC_50_ 0.04 mg/mL	[54]
		EC_50_ 2.6 mg/mL	[55]
Ferric reducing power (FRAP)	Methanol extract	3.69 ± 0.03 mg/mL Revealed antioxidant properties EC_50_ < 7.5 mg/mL	[51]
		31 µmol Fe^3+^ Eq/g d.w.	[53]
		5.5682 ± 1.0493 AAE/g	[54]
		Abs_700nm_ 0.08–0.35	[54]
	Ethanol extract	4.2750 ± 0.2629 AAE/g	[54]
		Abs_700nm_ 0.06–0.3	[54]
	Water extract	17.3972 ± 1.7329 AAE/g	[54]
		Abs_700nm_ 0.16–0.65	[54]
		Abs_700nm_ 0.03–0.89 for 0.625–10 mg/mL, respectively Abs_700nm_ 1.5–2.0 for ascorbic acid (0.625 and ≥1.25 mg/mL)	[55]
	Acetone extract	Abs_700 nm_ 0.1 ± 0.0, 0.1 ± 0.0 and 0.1 ± 0.0 for 0.5, 1 and 2 mg/mL, respectively Abs _700 nm_ 3.9 ± 0.9, 2.1 ± 0.0, 1.6 ± 0.0 for ascorbic acid (0.5, 1 and 2 mg/mL, respectively)	[56]
Copper reducing power (CUPRAC)	Water extract	Abs_450nm_ 0.14–0.81 for 0.625–10 mg/mL, respectively Abs_450nm_ 0.9–4.1 for BHT (0.625–10 mg/mL) and 0.44–2.1 for α-tocopherol (0.625–10 mg/mL)	[55]
Ferrous metal chelating activity	Methanol extract	66.59 ± 0.7418 Na_2_-EDTA Eq/g	[54]
		IC_50_ 1.1 mg/mL	[54]
	Ethanol extract	35.1287 ± 0.8974 Na_2_-EDTA Eq/g	[54]
		IC_50_ 2.0 mg/mL	[54]
	Water extract	40.46 ± 0.3143 Na_2_-EDTA Eq/g	[54]
		IC_50_ 1.8 mg/mL	[54]
Superoxide anion scavenging	Acetone extract	IC_50_ 221.8 ± 3.1 µg/mL Ascorbic acid 115.6 ± 1.2 µg/mL	[56]
Inhibition of β-carotene bleaching	Methanol extract	EC_50_ 1.79 ± 0.51 mg/mL Revealed antioxidant properties EC_50_ < 7.5 mg/mL	[51]
Inhibition of linoleic acid oxidation	Methanol extract	70%	[36]
	Water extract	10 mg/mL—70%, 6 mg/mL—10%	[55]
Extract of polysaccharides			
DPPH	Methanol extract	EC_50_ 0.10 mg/mL	[57]
ABTS	Methanol extract	EC_50_ 0.15 mg/mL	[57]

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
