# Peer review of "Black Trumpet [Craterellus cornucopioides (L.) Pers.]—Bioactive Properties and Prospects for Application in Medicine and Production of Health-Promoting Food"

_nutrients, 2024, doi:10.3390/nu16091325_

Round 1

Reviewer 1 Report

Comments and Suggestions for Authors

General consideration about the manuscript:

The authors discussed in detail chemical composition and functional activities of Craterellus cornucopioides. It’s not clear why the authors actually choose Craterellus cornucopioides. They should highlight any peculiarities of this mushroom compared, for example, to other species of the genus or to other edible mushrooms.

Moreover, a recent review (Bumbu et al. 2024 Nutrients 2024,16, 831) published in the same journal, not cited nor discussed by the authors, covers the same topics. I suggest the authors to cite this review and highlight any novelty of their work.

Please improve images resolution throughout the manuscript.

Further comments to the manuscript:

Page 2, lines 57-68: please specify if the chemical composition reported referes to dry weight or fresh weight. This is specified only for some nutrients. I suggest to report all the results to dry weight, in order to simplify reading as well as comparison of the results. Please specify dry weight or fresh weight troughout tha paragraph “Chemical composition”

Page 2, lines 59-60 and Figure 2: why is the fat content reported by Ouali et al. (2023) so high?

Page 3, lines 78-81: to which Reference/s do these results refer?

Page 3, lines 88-89: “They accounted for 83.6% and almost 60% and 75.9% and 61.4% ” This sentence is not clear. Did the author mean that the range of UFA and MUFA wer 60.0-83.6% and 61.4-75.9%, respectively? Please better explain. Are these results on fresh or dry weight? The same for results from lines 91-99.

Page 3, lines 72-85: Carbohydrates generally represent the highest component (on a dry weight) in edible mushrooms. Please discuss more in detail this nutrient.

Page 3, line 104: according to Table 1, the highest amount of potassium was reported by Ouali et al. [15], not by Turfan et al. [17]. Please verify.

Page 4, line 113: what did the authors mean with mg/100g ww?

Page 4, lines 125-135: Please specify if the results resfer to dry weight or fresh weight. Why are the results reported by Radovic et al. [12] so high?

Page 5, lines 142-146: Please give more importance to ergosterol and ergosterol derivatives in mushrooms, due to their biological functions and precursor of vitamin D2, as well as to their general high amount in edible mushrooms. Other works are present in the literature regardin ergosterol in Craterellus cornucopioides: Villares et al. 2014 (Food Chem. 147, 252–256), and Dospatliev et al. 2023 ( Journal of microbiology, biotechnology and food sciences 12.6: e4718-e4718) are only some examples. How do the authors explain the presence of cholecalciferol in Craterellus cornucopioides reported by Radović  et  al. 2022? As stated by the same authors in their work, “There is no literature data that support this founding”. It’s well known that cholecalciferol, or vitamin D3, originates from animal sources, while vitamin D2 (ergocalciferol) comes mainly from vegetable sources, and edible mushrooms are generally considered a good source of Vitamin D2 ((Ritota and Manzi 2023, AIMS Agriculture and Food, 8(2): 391-439)

Page 5, lines 167-169-176: Please specify if the results resfer to dry weight or fresh weight.

Page 6, line 214: Please correct the Reference “Arora 1991 after [2]

Page 7, line 251: Please correct the Reference “Belitz et al. 2009 after [36].”

Page 7, lines 251-253: Also 5’-nucleotides are responsible for the umami taste of edible mushrooms (Ritota and Manzi 2023, AIMS Agriculture and Food, 8(2): 391-439), as specified by the authors in page 8, lines 288-291.

Page 8, lines 291-292: Please correct the Reference “Chen 1986 after: [11]”

Comments on the Quality of English Language

Extensive editing of English language required

Reviewer 2 Report

Comments and Suggestions for Authors

the structure of the manuscript is not in accordance with the instructions of the MDPI publishing house, edit the individual subsections (marked with numbers) according to the MDPI template

What is your own scientific experience with this mushroom?

Line 33 - In what sense, according to your knowledge, does it change the sensory value of food?

Figure 3 - Is this your own picture?

the exact source data should be given under the legend of each figure

Line 415 - all latin names must be in italic (e.g. Klebsiella pneumoniae)

Round 2

Reviewer 1 Report

Comments and Suggestions for Authors

The authors replied to all Reviewer's comments and significantly improved the quality of their manuscript